# Unravelling the Transcriptional Response of *Agaricus bisporus* under *Lecanicillium fungicola* Infection

**DOI:** 10.3390/ijms25021283

**Published:** 2024-01-20

**Authors:** Luis Felipe Quiroz, Tessa Ciosek, Helen Grogan, Peter C. McKeown, Charles Spillane, Galina Brychkova

**Affiliations:** 1Agriculture and Bioeconomy Research Centre, Ryan Institute, University of Galway, University Road, H91 REW4 Galway, Ireland; luis.quiroz@universityofgalway.ie (L.F.Q.); charles.spillane@universityofgalway.ie (C.S.); 2Teagasc, Horticulture Development Department, Ashtown Research Centre, D15 KN3K Dublin, Ireland; helen.grogan@teagasc.ie

**Keywords:** dry bubble disease, cell death, ROS network, cell proliferation, button mushroom

## Abstract

Mushrooms are a nutritionally rich and sustainably-produced food with a growing global market. *Agaricus bisporus* accounts for 11% of the total world mushroom production and it is the dominant species cultivated in Europe. It faces threats from pathogens that cause important production losses, including the mycoparasite *Lecanicillium fungicola*, the causative agent of dry bubble disease. Through quantitative real-time polymerase chain reaction (qRT-PCR), we determine the impact of *L. fungicola* infection on the transcription patterns of *A. bisporus* genes involved in key cellular processes. Notably, genes related to cell division, fruiting body development, and apoptosis exhibit dynamic transcriptional changes in response to infection. Furthermore, *A. bisporus* infected with *L. fungicola* were found to accumulate increased levels of reactive oxygen species (ROS). Interestingly, the transcription levels of genes involved in the production and scavenging mechanisms of ROS were also increased, suggesting the involvement of changes to ROS homeostasis in response to *L. fungicola* infection. These findings identify potential links between enhanced cell proliferation, impaired fruiting body development, and ROS-mediated defence strategies during the *A. bisporus* (host)–*L. fungicola* (pathogen) interaction, and offer avenues for innovative disease control strategies and improved understanding of fungal pathogenesis.

## 1. Introduction

Mushrooms are nutritious crops with a very low carbon footprint of 0.07–0.14 kg CO_2_e/kg during cultivation [1,2]. The white button mushroom, *Agaricus bisporus*, is among the four most widely grown edible mushroom species. They are cultivated and consumed in over 70 countries, contributing 11% of the world’s mushroom supply in 2018-2019 [3]. *A. bisporus* is, however, susceptible to bacterial and fungal pathogens that can impact the growth, appearance, and overall quality of the mushroom, and which pose a major economic concern for mushroom growers worldwide [4]. In Western countries, the average annual damage due to pathogenic fungi affecting mushroom cultivation was estimated to be 25% of the total production value [5]. Control strategies are limited, as both the host and pathogen are fungi, making fungicide application challenging, and there is limited understanding of disease pathogenesis. *Lecanicillium fungicola* is a mycoparasite responsible for dry bubble disease in commercially cultivated mushrooms (Figure 1), predominantly affecting *A. bisporus* [4]. This parasitic disease is characterized by the formation of small cinnamon-brown necrotic lesions on the fruiting body caps of mushrooms if the infection starts at later developmental stages, or partial deformation of the fruiting bodies, including the formation of undifferentiated masses of tissue or ‘dry bubbles’, if infection occurs earlier (Figure 1 and [6,7]). The resistance of *L. fungicola* to some antifungal treatments raise concerns about the impact of this pathogen on global mushroom production in the future [3,4,6,8].

The differential response of *A. bisporus* to pathogens at early and late developmental stages is intricately linked to the chemical composition and structure of host cell walls, notably involving lectins in button mushrooms (ABA, *A. bisporus* agglutinin) [9]. ABA plays a pivotal role in recognizing and reversely binding to glucogalactomannan produced by *L. fungicola*, serving as a potential defence mechanism against *L. fungicola* infection. Upon germination of *L. fungicola* on developed *A. bisporus* fruiting bodies, *A. bisporus* orchestrates a downregulation of lectin production on hyphal surfaces [10]. Concurrently, the mushroom induces the production of chitin deacetylase, cell-wall-degrading enzymes, and oxidoreductases in response to pathogen attack [11], resulting in the alteration of the surface composition of *A. bisporus* mycelia. This biochemical response impedes the ability of *L. fungicola* hyphae to degrade the cell walls of *A. bisporus* vegetative mycelium, leading exclusively to the formation of cinnamon-brown cap lesions as a reaction to infection, possibly triggered by secondary metabolites [12].

These lesions exhibit similarities to the hypersensitive response (HR) observed in plants and are associated with resistance to the pathogen [13]. Scanning electron micrographs validate that during attacks on fruiting bodies, *L. fungicola* hyphae can only adhere to the surface of the vegetative mycelium or roll around *A. bisporus* mycelium [14]. This adhesion is facilitated through hydrophobic interactions between hydrophobin on the surface of *L. fungicola* hyphae and ABH3-type hydrophobin on the surface of vegetative mycelium of *A. bisporus* [9]. These adhesion mechanisms enable *L. fungicola* to persist when new mushroom primordia begin to develop. Indeed, a metagenomic analysis confirmed that *L. fungicola* can be present in compost, casing, and asymptomatic fruitbodies of *A. bisporus*, and the relative abundance of pathogenic fungi exponentially increased with the age of the crops [15]. It is suggested that nutrient leakage (likely carbon) from white button mushroom hyphae stimulate *L. fungicola* germination and growth toward the source of nutrients, such as the young fruiting bodies or mushroom pins [16]. In contrast to *A. bisporus* vegetative hyphae, the hyphae of young fruiting bodies are not resistant to the lytic enzymes of *L. fungicola* [9], which leads to primary infection and the appearance of undifferentiated tissues. The exact cellular and molecular mechanisms behind the pathogenesis of dry bubble disease remain largely unknown, hindering the development of effective preventive measures, despite some light being shed on the interaction between *L. fungicola* and *A. bisporus* during secondary infection [11].

*A. bisporus*, like other mushroom-forming fungi belonging to *Agaricomycetes*, has a mechanism of fruiting body formation that is divided between the early phase of cell proliferation and differentiation and the growth phase. *A. bisporus* undergoes morphogenetic cell death in developing primordia prior to basidial differentiation, leading to correct cellular patterning as the mushroom develops [17]. The occurrence of morphogenetic cell death was confirmed through protein ubiquitylation [18], and by observing the expression pattern of E3 ubiquitin ligase complex genes (linked to protein degradation) and cell cycle regulation genes expressed at specific morphological stages of fruiting body development [19,20]. However, when *L. fungicola* attacks button mushrooms at the primordia stage, the morphological disturbance leads to formation of undifferentiated mycelia masses, likely linked with disturbance of the cell cycle dynamics [9]. Transcriptomic analysis of the interactions between *A. bisporus* and *L. fungicola* also revealed upregulation of ROS network genes, like NADH-ubiquinone oxidoreductase, and downregulation of cell division control proteins during secondary infection [11]. The abovementioned suggest the possible involvement of the ROS network, cell cycle regulation, and apoptosis-related pathways in dry bubble disease pathogenesis. To test this hypothesis, we studied primary dry bubble disease pathogenesis by investigating the response of ROS network genes and those involved in fruiting body development, cell cycle regulation, and apoptosis during “bubble” formation. Identifying potential molecular or cellular targets could open possibilities for innovative antifungal strategies, such as developing protective methods for mushroom cultivation or the creation of new mushroom varieties. Hence, we quantify the transcriptional response of *A. bisporus* to dry bubble disease, shedding light on the possible mechanisms governing the mushroom’s defence strategy.

## 2. Results

### 2.1. L. fungicola Infection Affects the Transcription of Genes Involved in the Control of Cell Division, Fruiting Body Development, and Apoptosis

To determine in more detail the response of *Agaricus bisporus* to *L. fungicola* infection, mycelial samples were infected with spore suspensions of two *L. fungicola* strains: strain 1722, which is mildly pathogenic, and strain 620, which is strongly pathogenic. Both samples displayed symptoms of disease, with the expected difference in post-infection severity being observed between the mildly and strongly pathogenic strains (Figure 2a). Nearly 80% of mushrooms were scored “severely damaged” in the case of infection with the aggressive strain 620. In contrast, in the case of infection with the mild strain, 20 to 70% were damaged: absence of mushroom-body formation (scored as “mildly damaged” was observed in half of the cases, while other symptoms ranged from “not severely damaged” when lesion spots were noticeable (circa 30% of damaged mushrooms) to “severely damaged” when symptoms resembled those caused by the more aggressive strain. The primary infestation of white button mushrooms with *L. fungicola* spores at an early developmental stage leads to abnormal mushroom tissue development and prevents *A. bisporus* cell differentiation, irrespective of the aggressiveness of the *Lecanicillium* strain involved (Figure 2a).

This indicates that early infection is able to disrupt *A. bisporus* cell cycle regulation, fruiting body development, and, potentially, also apoptosis (given the role of cell death in mushroom development described above). We therefore performed a BLAST search against the reference *A. bisporus* var. *bisporus* H97 genome and identified putative sequences of genes which are involved in these processes (Appendix A). To verify primer specificity, BLAST analyses were conducted against the available *L. fungicola* genome in the NCBI database (GenBank: GCA_900169235.1), despite the absence of gene annotations. Additionally, comprehensive primer blast analyses were performed against the complete RefSeq mRNA database on NCBI, spanning various organisms. This validation process ensures the specificity and suitability of the primers for targeted amplification in our experimental procedures.

After confirming that primers specifically bind only with *Agaricus* cDNA, but not with the *L. fungicola* genome, the transcriptional responses of *A. bisporus* to infection were then identified. After 8 days post-infestation with *L. fungicola* spores, very few mushrooms with developmental abnormalities were observed (8 days, first flush); however, 10 days after infection, the dry bubble symptoms started to develop. On day 14, clear differences in disease severity were observed between strain 1722 and strain 620 (Figure 2a). Therefore, a real-time qPCR analysis was performed using “mildly-damaged” infected mushrooms from the second flush (14 days post-infection), and non-infected mushrooms of the same developmental stage were used as a control.

The differential transcriptional responses of putative orthologs to genes involved in cellular division, such as *septin-like 1* (*SEPT1*), *SEPT2* [21], cellulose-growth-specific *1* (*CEL1*) [22,23,24], *RAB domain-containing cell division control protein* (*RabCDC*) [25], *cell division control protein 5* (*CDC5*), and *cell division control protein 7* (*CDC7*) [26], were recorded (Figure 2b). However, differences between the responses to different strengths of strains infection were also observed. For instance, after infection with the aggressive *L. fungicola* strain, transcript levels of *SEPT1/2* and *RabCDC* were upregulated 5-fold, but were increased by more than 15-fold after infection with the mild strain (Figure 2b). Likewise, putative *A. bisporus* orthologs to *fruit-body specific A* (*FBSA*), *FBSB*, and *FBSC* [27] increased less after infection with the strong strain compared to the mild one (Figure 2c). Additionally, when analysing putative orthologs to genes related to apoptosis, such as *BAX inhibitor-1* (*BI-1*) [28] and *Laccase 10* (*LAC10*) and *LAC11* [29], only *BI-1* and *LAC10* were significantly increased in both infection treatments. However, while in mushrooms infected with the strong strain, *BI-1* transcript levels were lower than in mild infection, the *LAC10* transcript levels were equally high, with up to 300-fold increases (Figure 2d).

### 2.2. L. fungicola Infection Produces Impaired Reactive Oxygen Species (ROS) Levels in A. bisporus

At cellular and molecular levels, ROS act as key signalling molecules [30]. However, an overproduction of ROS can also lead to oxidative stress, producing damage to proteins, lipids, and DNA [31,32,33]. Through DAB staining, it was observed that infection with the strong *L. fungicola* variant induces a significant increase in hydrogen peroxide (H_2_O_2_) content in *A. bisporus*. In the case of infection with the mild variant of *L. fungicola*, despite no significant increase in H_2_O_2_ being observed, a tendency of increase can be noticed (Figure 3a).

Furthermore, the transcript levels of putative orthologs to genes encoding proteins involved in ROS production (Figure 3b), such as *NADH-ubiquinone oxidoreductase subunit (NUXM*) [34], *ferric reductase NAD binding domain-containing protein* (*FRN*) [35], *polyphenol oxidase 3* (*PPO3*), and *PPO4* [36], as well as putative genes involved in ROS scavenging [37,38], such as superoxide dismutase (*SOD*), *catalase* (*CAT*), *glutathione reductase* (*GR*) and *thioredoxin reductase* (*TR*), and *urea amidohydrolase subunit alpha* (*UREA*), were analysed. It was observed that relative transcript levels of *NUXM*, *PPO3,* and *PPO4* were increased in both mild- and strong-strain infections (Figure 3b). However, strong-strain infection produced lower induction; in the case of ROS-scavenging genes, while transcript levels of *SOD*, *GR*, *UREA*, and *CAT* were elevated after mild infection, no increase was observed for *SOD* and *GR*, and a slight decrease was actually observed in *UREA* transcript levels following aggressive infection (Figure 3c). A slight, but not statistically significant, decrease was observed in *TR* in either condition compared to control (Figure 3c). This indicates that a differential response can be observed at the molecular level depending upon the severity of the infection, with the more aggressive strain tending to induce fewer putative response mechanisms in the *A. bisporus* host.

## 3. Discussion

Understanding the molecular mechanisms of dry bubble disease development will contribute to future disease treatment and prevention. Moreover, gaining insights into the pathways involved in the dysregulation of the cell differentiation process in mushrooms could provide valuable knowledge for understanding the pathogenesis of similar processes in other organisms and diseases. To date, the focus has been on understanding the interaction between *A. bisporus* and L. *fungicola* at the secondary infection stage to identify mechanisms of infection prevention [11,16]. This study investigates the transcriptional response of *A. bisporus* genes involved in different cellular and molecular processes during the initiation of the primary infection with mild or strong *L. fungicola* strains. Coupled with the nature of the “dry bubble” phenotype, wherein amorphic, cancer-like tissue is produced (Figure 2a), these results lead us to propose a model in which the processes of cell differentiation and cell death are dysregulated (Figure 4).

SEPT proteins, which play crucial roles in cell division as components of the contractile ring at the cleavage furrow, facilitating the physical separation of daughter cells during cytokinesis [21], were elevated 5- to 15-fold in response to infection (Figure 2b). Similar upregulation was noticed for CDC5, a Polo-like kinase in yeast which orchestrates mitotic entry and acts as a positive regulator of cell cycle G2/M progression by activating key cell cycle regulators, thus ensuring proper spindle formation and DNA replication [39,40,41]. Although there is no direct homolog in humans, *POLO-LIKE KINASE 1* (*PLK1*) is a related gene which shares functional similarities, the over-expression of which is associated with various kinds of cancer [42,43,44]. Thus, the increased *SEPT1/2* and *CDC5* transcript levels could suggest an enhanced cell division process, potentially contributing to abnormal tissue growth and development in the host mushroom. The dysregulation of cell division control genes, especially in the context of CDC5, may have implications for pathological conditions, potentially leading to abnormal tissue growth or disease development. This study suggests a connection between the upregulation of these genes and the potential development of necrotic or cancer-like diseases, highlighting the importance of cell division control in fungal morphogenesis and pathogenesis.

*FBS* genes play a crucial role in the development of fruiting bodies in mushroom-forming fungi. These genes are involved in various functions such as sexual development, basidium formation, and sporulation [45]. These fruiting bodies are among the most complex structures produced by fungi. Unlike vegetative hyphae, fruiting bodies grow determinately and follow a genetically encoded developmental program that orchestrates tissue differentiation, growth, and sexual sporulation [46]. Hence, highly upregulated *FBSA* and *FBSC* transcripts under *L. fungicola* infection, allow us to suggest the potential involvement of these genes in impaired fruiting body development. This heightened expression indicates the initiation of fruiting body development as a response to the pathogen, potentially enhancing the reproductive efforts of *A. bisporus* for survival. Alternatively, *L. fungicola* might be exploiting the cellular machinery of the host to promote its own growth. By inducing fruiting body development, the pathogen could benefit from the resources and conditions such structures provide, potentially increasing its chances of propagation in the face of a pathogenic threat.

Cells often employ PCD in response to external challenges, to prevent damage at the tissue or organism level. Interestingly, BI-1, which encodes for an anti-apoptotic protein that indirectly inhibits the pro-apoptotic factor BAX [47], was transcriptionally induced under infection with *L. fungicola* (Figure 2d). BI-1 is a highly conserved endoplasmic reticulum transmembrane protein which also regulates endoplasmic reticulum stress responses, calcium imbalance, and ROS homeostasis, with potential implications in disease contexts [48,49]. Central to these processes are ROS, which play a pivotal role as signalling molecules in cellular division, development, and PCD regulation [30,38,50,51,52,53]. ROS helps to derive the alterations and reprogramming of cells to modulate the expression of signalling components and pathways associated with cell growth and homeostasis [54].

ROS homeostasis plays a central role in cellular defence mechanisms in response to endogenous and exogenous stimuli in many organisms (Figure 4 and [55]). Highly reactive superoxide anions are rapidly converted by superoxide dismutase (SOD) into H_2_O_2_, and indeed, after infection with the mild strain, a rapid increase in SOD transcript level was noticed (Figure 3c). Similarly, a 1.5-fold increase in transcript levels of Glutathione Reductase (GR), which is crucial for H_2_O_2_ elimination via the ascorbate–glutathione cycle, was noticed in mushrooms infected with the mild strain. We also noticed 6–8-fold increases in the urea amidolyase (UREA) enzyme, which catalyses two consequent reactions through two domains, urea carboxylase and allophanate hydrolase, and stimulates antioxidant metabolism [37,38]; the by-products of the reaction, CO_2_ and HCO_3_^-^, further modulate free radical and peroxide-mediated reactions [56].

Through mediation of different pathways, ROS determine whether cells undergo G1-S arrest or continue proliferating [57]. Likewise, by regulating growth factors, ROS mediate growth and development (e.g., FBSA, FBSB, FBSC genes; Figure 2c). This further leads to the activation of MAP kinase cascades, which play major roles in the regulation of proliferation, differentiation, immune and stress responses, transformation, and apoptosis [58]. ROS also mediate the intrinsic apoptosis pathway, affecting the expression and functions of pro- and anti-apoptotic proteins (Figure 2d) and thus enhancing the process of proteasomal degradation which finally leads to apoptotic death [54].

In this context, laccases, copper-containing enzymes, display a dual role in ROS dynamics. While these enzymes can produce ROS, during substrate oxidation, they also contribute significantly to ROS detoxification [59,60]. This dual functionality highlights the nuanced and essential role laccases play in managing ROS levels. Laccases are pivotal in mushroom biology, being involved in multiple processes spanning lignin degradation, vegetative and reproductive development, wound response, pathogen defence, and redox signalling [61]. Thus, the increase in LAC10 transcript levels (Figure 2d) might also be part of a mechanism to regulate ROS levels and oxidative stress. Intriguingly, in animal and cellular cancer models, laccases can induce or modulate apoptosis-related genes, somehow exerting antitumoral activity [62,63,64]. Thus, it could be hypothesized that LAC10 transcriptional increase could be a response to modulate the excessive and decontrolled cellular division in *A. bisporus* infected with *L. fungicola.*

Overall, the increase in the relative transcript levels of the ROS-related genes NUMX, PPO3, and PPO4 during the host–pathogen interaction between *A. bisporus* and *L. fungicola* could indicate the activation of a ROS-mediated defence response by the host [31,32,33]. However, it could also be possible that an overproduction of ROS as a protective response could lead to detrimental effects in the host, such as cellular damage and necrosis [31,65,66]. Nonetheless, to fully understand how the balance between ROS-mediated defence and potential oxidative stress is involved in host–pathogen response warrants further investigation to understand the full implications of these gene expression changes.

Interestingly, the smaller phenotypic effect and reduced ROS production under mild *L. fungicola* infection (Figure 2a and Figure 3a) correlate with the increase in transcript levels of genes encoding proteins involved in ROS scavenging, specifically SOD, GR, CAT, and UREA (Figure 3c). SOD plays a pivotal role by converting superoxide radicals into less harmful molecules [67], while GR ensures a continuous supply of reduced glutathione for antioxidant defence [68]. CAT complements this by efficiently breaking down hydrogen peroxide into molecular water and oxygen [69]. The possible collaborative action of these enzymes could form a robust antioxidant defence system, crucial for neutralizing ROS, maintaining cellular redox balance, and safeguarding cells from oxidative stress-induced damage. Together, these enzymes contribute to cellular health and resilience in the face of oxidative challenges. UREA is an enzyme that catalyses the hydrolysis of urea into ammonia and carbon dioxide. This enzyme is known to play a role in nitrogen metabolism and pH regulation. pH levels can be altered through ammonia production, creating a less favourable microenvironment for ROS production [70]. However, it is still not determined if this increase in ROS scavenging enzymes is induced for the pathogen as a mechanism to neutralize host-derived ROS or as a host response to counteract the detrimental effect of the pathogen [71]. Overall, these results provide important insights into the molecular aspects of the response of *A. bisporus* to the pathogen and set the stage for more in-depth investigations into the intricate interplay between ROS signalling, host defence, and pathogen adaptation.

## 4. Materials and Methods

### 4.1. A. bisporus Infection with L. fungicola Strains

A crop trial was set up and carried out in an environmentally-controlled mushroom growing room at the Mushroom Research Centre, Teagasc. The growing containers (25 cm diameter pots) were filled with 4.5 kg of commercial mushroom substrate and then covered with a 5 cm layer of peat-based mushroom casing soil. The containers were incubated at 25 °C for 11 days prior to inoculation with two isolates of *L. fungicola*: the 1722 isolate (mild strain) and the 620 isolate (strong, aggressive, strain). Three distinct treatments of primary *Agaricus* infestation with *L. fungicola* spores, each executed in quadruplicate, were performed. The first treatment comprised a control group, maintained uninoculated to establish a baseline. The second and third treatments involved the inoculation of the casing soil with isolates 620 and 1722, respectively, wherein a concentration of 1 × 10^6^ spores/m^2^ was applied. The spore suspensions of strains 1722 and 620 were obtained by adding 5 mL distilled water to the potato dextrose agar (PDA) plates with corresponding *L. fungicola* strains. *L. fungicola* cell counts were performed for each spore suspension, in haemocytometer chamber (5 sets of 16-corned squares per replication), to obtain an average concentration of 1 × 10^6^ cfu/mL. The spore suspension was next diluted in distilled water to a concentration of 0.4 × 10^4^ cfu/mL and 12.5 mL of spore suspensions were added to each plot to a final concentration of 1 × 10^6^ spores/m^2^, according to experiment layout. The inoculation was performed on day 11 after crop set up. The samples were collected starting from 14 days post-inoculation (2nd flush). Fruitbody samples were scored according to the symptom severity (1–5, least severe symptoms; 5–10, progressed symptoms; 10–15, necrotic symptoms). The fruitbody samples were collected at equal intervals post-infection from symptomatic mushrooms. Control samples were collected from mushrooms at similar developmental stages. For the current research, samples collected 14 days post-inoculation were used. All samples were stored at −80 °C for further analysis.

### 4.2. Primer Design

The primers were designed against *Agaricus bisporus* var. *bisporus* H97 [72]. Since cellular division, cell cycle regulation, and fruit body development are essential for mushroom-like body formation [19,20], the genes involved in these functions were selected. *Septin-like* genes (*SEP1* and *SEP2*) are involved in the cell division during cell cycle [21]. The *Cellulose-growth-specific 1* (*CEL1*) genes were shown to be involved in cell cycle progression in the G1 stage, but not at the G2/M stage, and are critical for the cell size checkpoint [24]. The RAB Domain-Containing Cell Division Control protein (Rab-CDC) could likely be the major regulator of cellular division [25] and cellular division control proteins were selected based on their role in the morphological appearance of mushrooms [26]. ROS network genes involved in ROS production: (*NADH-ubiquinone oxidoreductase subunit* (*NUXM*); *Ferric reductase NAD binding domain-containing protein* (*FRN*); Polyphenol oxidase (*PPO*) 3 and 4 (*PPO3* and *PPO4*)) and genes encoding enzymes involved in ROS scavenging: (*Superoxide dismutase* (*SOD*); *Glutathione reductase* (*GR*); *Urea amidohydrolase subunit alpha* (*UREA*); *Catalase* (*CAT*); *Thioredoxin reductase* (*TR*)) [34,38,73]. *β-tubulin 1* (*tubb1*) and *Elongation Factor-1a* (*EF1a*) were used as housekeeping genes [74,75]. Since they yielded similar results, the qRT-PCR results presented against tubb1 were used as a housekeeping gene.

### 4.3. RNA Extraction and Quantitative (qRT-PCR)

RNA was extracted from *A. bisporus* tissue samples using an Aurum Total RNA Mini Kit (Cat. 732-6820, Bio-Rad, Hercules, CA, USA) and following the protocol given by the provider. The concentration and purity of the extracted RNA were determined using a spectrophotometer (Nanodrop brand/model) by measuring the absorbance ratio at 260 nm/280 nm and 260 nm/230 nm, and RNA integrity was determined by gel electrophoresis. To synthesize cDNA, 1 μg of treated RNA was mixed with 1 mM oligo dT and PhotoscriptII reverse transcriptase (NewEnglands Biolabs, Ipswich, MA, USA). The RNA without PhotoscriptII reverse transcriptase was used as a negative control (cDNA minus). The quality of synthesized cDNA was verified by PCR with *β-tubulin 1* (*tubb1*) and *Elongation Factor-1α* (*EF1α*) primers (Appendix A) using MyTaq Red Mix (Bioline Ltd, London, UK). A total of 1 μL of each cDNA (plus/minus) was used in 25 μL PCR reaction using a standard 3-step cycling profile [95 °C—1 min; followed by 33 cycles of 95 °C for 15 s, 60 °C for 15 s, and 72 °C for 10 s, completed with 1 cycle at 72 °C for 2 min). The 10 μL of final product was run on 2% agarose gel with Sybr Safe DNA gel stain (Invitrogen, Carlsbad, MA, USA). The Real-Time PCR analysis was performed using a QuantStudio™ 5 Real-Time PCR System (Applied Biosystems, Beverly, MA, USA). The cDNA of all samples used for qRT-PRC analysis was normalized against *tubb1*, followed by dilution with water and 40× Yellow Sample buffer to the final concentration of 1×. The master mix was prepared according to the PowerTrackSYBR Green Master Mix manufacturer instructions (Applied Biosystems, Beverly, MA, USA). The qPCR was set using a standard 3-step cycling profile [95 °C—1 min, followed by 40 cycles of 95 °C for 15 s, 60 °C for 25 s, and 72 °C for 30 s), followed by standard melting curve settings (95 °C for 15 s, 60 °C for 1 min, and 95 °C for 15 s). The relative expression of each gene was normalized against the β-tubulin gene (Appendix A).

### 4.4. A. bisporus DAB Staining

The “mildly-damaged” mushrooms infected with 620 and 1722 strains, and corresponding control samples, were collected 15 days post-infection and immediately used for ROS analysis. The mushroom slices (thickness up to 1 mm) were prepared with razor blades, directly into 20 mL of 50 mM Tris-HCl, pH 7.5. The mushroom slices were vacuum-infiltrated for 3 min and stained with 3,3′-diaminobenzidine reagent (DAB) 1 mg/mL, pH 5.0 [43]. Next, 5 h later, the mushroom slices were fixed with a solution of 3:1:1 ethanol/lactic acid/glycerol and photographed. To plot DAB staining levels (relative hydrogen peroxide), a false-coloured image was made using ImageJ, and the percentage of the area of each mushroom slice corresponding to pixels between 170 and 255 were counted as described before [38].

### 4.5. Data Analysis and Visualization

Statistical analysis was carried out using SPSS 28.0 (IBM Corp., Armonk, NY, USA. Released 2021. IBM SPSS Statistics for Mac, Version 28.0). The multiple comparison of means was performed by ANOVA followed by Tukey’s post hoc test at a confidence level of 0.95. Graphics were made using GraphPad 10.0 (GraphPad Prism Software for Mac, Version 10.0, San Diego, CA, USA). Schemes were created with Biorender.com (accessed on 10 January 2024).

## 5. Conclusions

The exploration of *A. bisporus* gene expression during *L. fungicola* infection unveils intriguing aspects of the host–pathogen interplay. The transcriptional upregulation of genes involved in cell division control, apoptosis, and fruiting body development hints at the idea of impaired cell division and structure development in infected mushrooms. Elevated ROS content and induced expression of genes encoding for ROS producing enzymes (NUXM, PPO3, and PPO4) as well as ROS scavenging (SOD, GR, UREA, and CAT), highlight the prime role of ROS in the response to *L. fungicola* infection. Thus, these findings offer insights into *A. bisporus* responses to *L. fungicola* infection, illustrating the broad transcriptional response and possible interconnected dynamics between ROS production/scavenging, cell proliferation, tissue differentiation, and apoptosis. However, more detailed functional studies, examining the samples immediately after pinning throughout the appearance of the dry bubble symptoms on transcriptome, proteome, and metabolome levels, are essential to unravelling the precise mechanisms, enriching our understanding of host defence strategies and disease control.

## Figures and Tables

**Figure 1 ijms-25-01283-f001:**
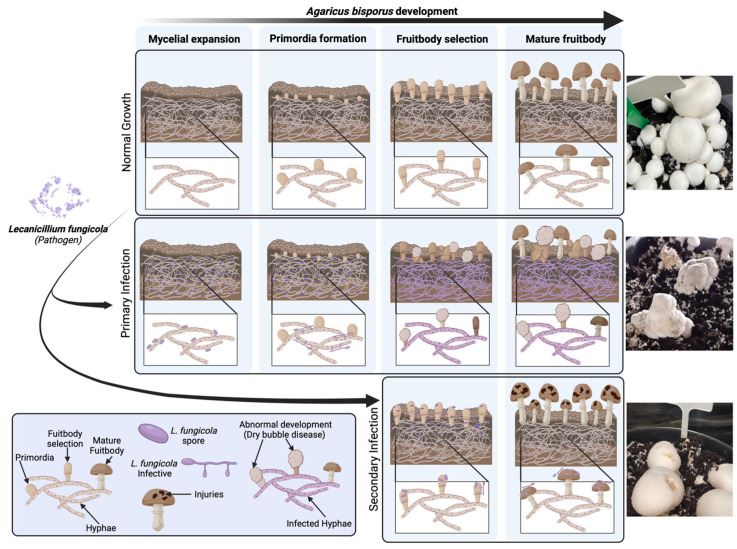
*Lecanicillium fungicola* infection in *Agaricus bisporus*. Disease symptoms depend on the life stage of the mushroom at which infection occurs. Abnormally shaped mushrooms, referred to as “dry bubbles”, form when mushrooms are infected at the very early stages of development (primary infection). Otherwise, infection in the late stages of development only causes injuries to the fruiting body (secondary infection).

**Figure 2 ijms-25-01283-f002:**
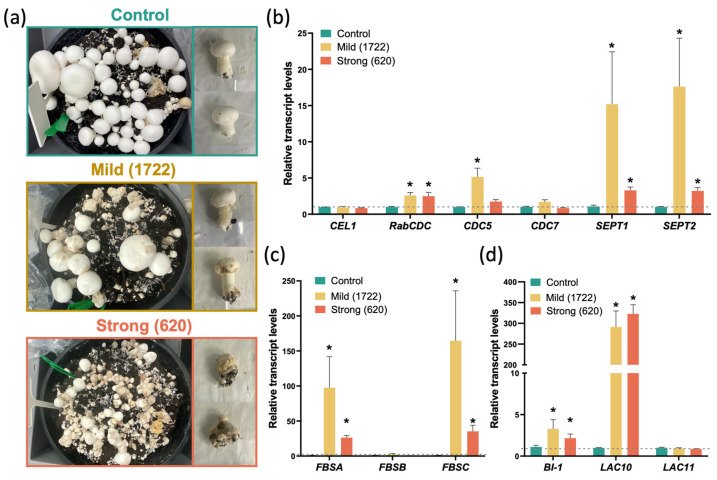
Transcriptional response of *A. bisporus* 14 days post-infection with *L. fungicola*. (**a**) Phenotypic analysis of mushrooms infected with mild and strong *L. fungicola* strains. (**b**) Relative transcript levels of genes encoding for proteins involved in cellular division. (**c**) Relative transcript levels of *FRUIT BODY-SPECIFIC* (FBS) genes. (**d**) Relative transcript levels of genes encoding for proteins involved in apoptosis. Transcript abundance was normalized to tubulin transcript levels and calibrated to control condition (no infection). All values represent the means of three independent replicates (+SD). Statistically significant differences were determined by two-tailed ANOVA test: * *p* < 0.05.

**Figure 3 ijms-25-01283-f003:**
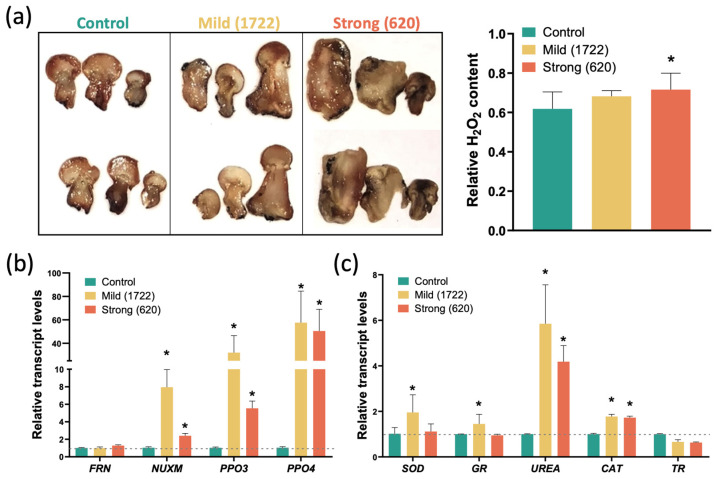
ROS-related response of *A bisporus* 14 days post-infection with *L. fungicola* infection. (**a**) DAB staining for hydrogen peroxide detection (left) and relative quantification (right). (**b**) Relative transcript levels of genes encoding for ROS producer enzymes. (**c**) Relative transcript levels of genes encoding for ROS scavenging enzymes. Transcript abundance was normalized to tubulin transcript levels and calibrated to control condition (no infection). All values represent the means of three independent replicates (+SD). Statistically significant differences were determined by two-tailed ANOVA test: * *p* < 0.05.

**Figure 4 ijms-25-01283-f004:**
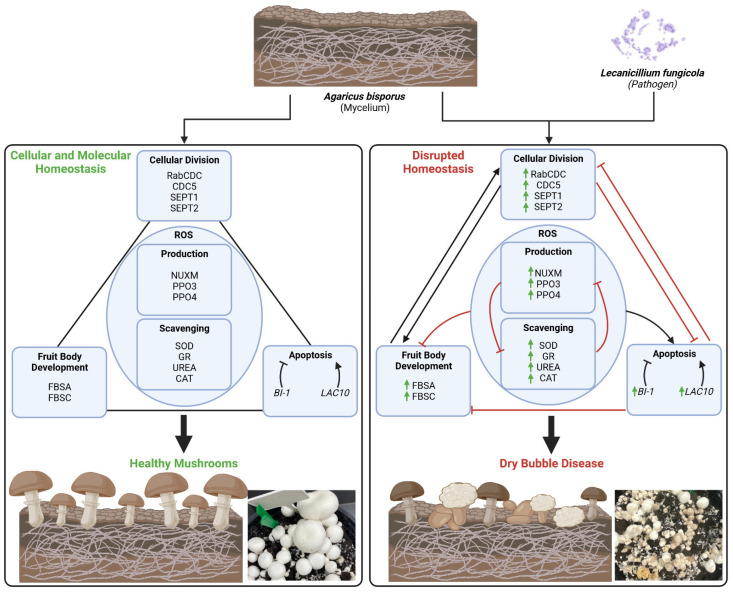
The possible mechanisms underlying dry bubble disease pathogenesis. *Lecanicillium fungicola* infection triggers ROS homeostasis dysregulation as a primary response to infection. When infestation happens at the early mushroom developmental stages, genes involved in cellular division, fruit body development, and apoptosis are dysregulated, resulting in the “dry bubble” phenotype.

## Data Availability

The data generated during the study are available upon request.

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
