# Peer review of "Unravelling the Transcriptional Response of *Agaricus bisporus* under *Lecanicillium fungicola* Infection"

_ijms, 2024, doi:10.3390/ijms25021283_

Round 1
Reviewer 1 Report
Comments and Suggestions for Authors
You must verify that the primers used only amplify the fungal pathogen targets, not those of the host fungus.
You must include the time -days, hours, or whatever- between inoculation and sampling. I a time course was used, that should be reflected in the results.

See suggestions on returned pdf.
Author Response
Dear Reviewer 1,
Thank you very much for investing your time and providing valuable comments, that significantly improved our manuscript. Please see detailed comments in the attached file.
Galina

Reviewer 2 Report
Comments and Suggestions for Authors
Overall conclusions: In this manuscript, Quiroz et al. conduct a qPCR based transcriptional study of specific gene targets in A. bisporus and how their expression is affected by A. bisporus and L. fungicola pathogenic interactions. The study provides some comparison between the transcriptional impact due to interactions with a strong vs. mild pathovar of L. fungicola. However, the study provides a highly narrow set of data about a small subset of gene targets and their expression at one timepoint during the pathogenic interaction, which makes it difficult to draw conclusions about the regulatory patterns. Issues with sampling depth and the setup of the experiment as well as data interpretation are mentioned in the specific comments.
Specific comments:
· Intro Line 56-58: Highlight the reasons why the previous studies have failed to understand the molecular interactions between the host and the pathogen, was it due to null results or due to methodological challenges? Further, mention how this study specifically addresses that.
· The results/methods section should include detailed reasoning behind the choice of all the specific gene targets evaluated by qPCR. This should include some previous citations about some of these genes evaluated in other studies. It seems unclear why only a specific set of genes were chosen.
· Method 4.3: The endogenous control should be clearly mentioned in the method section for qPCR.
· Method 4.1: It is unclear exactly how long after inoculation the infected A. bisporus samples were harvested. The description says that they were collected at similar times, but not specifically how long after infection. This would need to be consistent, otherwise the comparisons of the qPCR results from the samples are not correct.
· Figure 2 &3: From the methods, it is unclear how long the host and the pathogen had been interacting before the samples were collected for RNA. Further, it seems that only one timepoint or stage was sampled for all qPCR throughout the study. This is fine in terms of comparing the transcriptional effect of some genes due to the strong vs. mild L. fungicola strain, but it does not provide much information about the transcriptional response of A. bisporus during different stages of interaction with L. fungicola. In order to understand this, the study has to include different timepoints of interaction between the two species and evaluate the expression of the same target genes. The variability in this regard can be introduced in terms of the length of interaction as well as the developmental stages during which interaction occurs.
· Method 4.3/Figure 3: The accurate procedure of relative H2O2 determination is not mentioned in the methods and needs to be added.
· In the discussion section, the functional description of FBS, SEPT and CDC gene targets highlight the involvement of these genes/groups of genes during various stages of growth and differentiation. However, the data collected in this study is limited to one timepoint in this cycle which is not clearly outlined. Thus, it is difficult to draw conclusions about the impact of A. bisporus and L. fungicola on the transcription of these genes as it would vary over time and that data is not present here. In contrast, the study could also benefit from an RNA-Seq type approach that provides global transcriptomic information about the interaction at a particular timepoint or across timepoints.
· The lack of correlation between ROS gene target expression and peroxide concentration should be explained in the discussion section.
Author Response
Dear Reviewer 2,
Thank you very much for investing your time and providing valuable comments, that significantly improved our manuscript. Please see detailed comments in the attached file.
Kind regards,
Galina

Round 2
Reviewer 2 Report
Comments and Suggestions for Authors
Comments were addressed.